# Chronic Atherothrombosis in a Sub-Massive Infrarenal Abdominal Aortic Aneurysm in a 91-Year-Old White Male Donor

**DOI:** 10.3390/diagnostics12102270

**Published:** 2022-09-20

**Authors:** Andrea Sparks, Scott Oplinger, Elizabeth Maynes, Keiko Meshida, Maria Ximena Leighton, Gary Wind, Guinevere Granite

**Affiliations:** 1F. Edward Hebert School of Medicine, Uniformed Services University of the Health Sciences, Bethesda, MD 20814, USA; 2Department of Surgery, Uniformed Services University of the Health Sciences, Bethesda, MD 20814, USA; 3The Henry M. Jackson Foundation for the Advancement of Military Medicine, Inc., Bethesda, MD 20814, USA

**Keywords:** abdominal aortic aneurysm, atherosclerosis, atherothrombosis, ultrasound screening, open aneurysm repair, endovascular aneurysm repair

## Abstract

Screening for abdominal aortic aneurysms became the standard of care in 2005, yet screening procedures continue to be underutilized. While improvements in mortality rates have been noted over the past 15 years, continued patient mortality from ruptured abdominal aortic aneurysms suggests a need for further research, regarding the effectiveness of the current screening process. Abdominal aortic aneurysms can progress silently, and the risk of rupture increases significantly with increase in diameter. We report a large, untreated infrarenal abdominal aortic aneurysm of 17 cm in length and 8 cm in diameter, showing the chronic atherothrombosis discovered in a 91 year-old white male cadaveric donor. A literature review was conducted to elucidate current understanding of the pathology, risk factors, screening recommendations, and treatment options available for abdominal aortic aneurysms.

## 1. Introduction

Ruptured abdominal aortic aneurysms (AAA) have a mortality rate of up to 90% [1] and caused 9904 deaths in the United States in 2019 [2], as well as an estimated 175,000 deaths globally [3]. AAAs are most commonly present in men between the ages of 50–84, and smoking has been identified as a significant risk factor for the development and rupture of AAAs [4]. Hyperlipidemia, leading to subsequent atherosclerosis, is also a major causative agent of AAA formation [5]. The donor discussed in this case was a 91 year-old male donor who smoked and had hypercholesterolemia, exhibiting all of the most significant risk factors for the development of a AAA. The pathological process of AAA development will be briefly described in six steps, including endothelial injury, formation of cholesterol plaque, oxidative stress exacerbation, weakening of the arterial wall, and, finally, dilation of the vessel [5]. Ultrasonography is the most sensitive and specific screening method. Both the U.S. Preventive Services Task Force (USPSTF) and Society of Vascular Surgeons (SVS) recommend screening for men over 65 with a history of tobacco use [4,6]. It is unclear whether or not the donor in this discussion was ever screened. The most effective treatment for ruptured and unruptured AAAs is surgery, classified by two main approaches: endovascular aneurysm repair (EVAR) and open surgical repair (OSR) or open aneurysm repair (OAR). Since AAA development is associated with lifestyle-related risk factors, such as smoking and hyperlipidemia, smoking cessation, regular exercise, and healthy diets should be advised by health care professionals; furthermore, AAA screening should be implemented based on protocols of the USPSTF and other governing bodies. When a AAA is detected, close monitoring, follow up, and management is necessary. The most effective way to prevent a AAA from forming is through adequate screening and minimizing preventable risk factors, such as smoking and hyperlipidemia.

## 2. Case Description

During routine anatomical dissection of fifty-eight human donors in the 2021–2022 first-year medical gross anatomy course and 2022 graduate nursing advanced anatomy course at the Uniformed Services University of the Health Sciences, a large AAA was discovered in a 91-year-old white male donor who died of coronary artery disease (CAD). The donor, a military veteran, had a history of cardiovascular disease, including myocardial infarction, hypertension, and triple-bypass surgery (1987), as well as two hernia repairs, head trauma, and lung cancer. He also had a history of tobacco use, though a pack-year calculation could not be obtained. While his chart indicated a history of AAA, it is uncertain whether he was diagnosed or screened for AAA while he was alive. The AAA discussed in this case was infrarenal, beginning immediately distal to the renal arteries, and measured 17 cm long with an internal diameter of 8.5 cm (Figure 1). Upon dissection, bleeding was found in the left paracolic gutter, which suggests that the aneurysm might have ruptured into the abdomen. The iliac arteries were tortuous and highly sclerotic, with little usable area for blood flow. The inferior mesenteric artery was patent, likely from collateral vessels.

## 3. Discussion

### 3.1. Pathophysiology of AAAs Caused by Underlying Atherosclerosis

An aortic aneurysm is a dilation of the aorta, where the diameter of the aorta equals or exceeds 3.0 cm. Aneurysms occurring in the abdominal aorta are most commonly caused by atherosclerosis. Atherosclerosis serves as the most common underlying cause of coronary artery disease and peripheral arterial disease (PAD) [7].

The destruction of the vascular media, due to atherosclerosis, is caused by chronic endothelial injury. Hyperlipidemia is the most causative agent of the progression of atherosclerosis, and its presence alone is substantial enough to initiate pathological aortic dilation [5]. Endothelial dysfunction follows endothelial injury. The vasculature experiences increased permeability, which then allows lipoprotein molecules to enter the subendothelial space. The lipoproteins are retained and oxidized, becoming proinflammatory and proatherogenic [5]. The atherogenic stimuli activates the endothelium, which then upregulates adhesion molecules (primarily VCAM-1) and chemokines and recruits T-cells and monocytes [5]. The recruited monocytes mature into macrophages in the vascular intima. The chemokines and adhesion molecules inhibit endothelial nitric oxide synthase (eNOS), thus inhibiting the athero-protective effects of nitric oxide (NO); NO promotes smooth muscle relaxation, known as endothelium-dependent vasodilation [8]. Following vascular dysfunction, a solid plaque is formed as the macrophages are activated and engulf lipids, forming fatty streaks in the vasculature, and collagen is deposited. Macrophages full of engulfed cholesterol esters are renamed foam cells, which are an indicator of early and late atherosclerotic lesions. As the macrophages continue to engulf cholesterol esters and experience apoptosis, and plaques with destabilizing lipid-rich cores are developed. Atherosclerotic plaques are frequently found in areas of oscillating flow/turbulence and low shear stress. The posterior wall of the infrarenal aorta matches these conditions and is a common area of atherosclerotic plaque [7]. Infrarenal atherothrombosis near the iliac bifurcation, such as featured in the case discussed (Figure 2), is particularly likely, due to hemodynamic conditions that favor the outward, radial transport of molecules and collision of cells between one another and the vessel wall [9]. 

Large areas of plaque are observable in Figure 3b and marked with an arrow. White areas of calcification are also noted, which is a common progression seen in atherosclerotic plaques [5] (Figure 3b). The renal arteries and right iliac artery were also highly sclerotic. As the plaque continues to grow, the artery is remodeled. When the remodeling results in a more constricted blood vessel, the flow-limiting potential is accentuated [5]. The plaque observed in Figure 3b is an example of constrictive remodeling, where the area of blood flow was drastically reduced, especially at the distal end of the aorta, near the bifurcation of the iliac arteries (Figure 3a,b).

Aneurysms occur as changes in arterial pressure damage and thin the wall of the vasculature, thereby degrading the extracellular matrix and vascular smooth muscle [9]. AAAs can fall into two categories of morphology: fusiform or saccular. Fusiform aneurysms, being more common in AAAs, occur when the entire circumference of the artery is involved in the dilation. Saccular aneurysms occur when only a portion of the circumference of the artery is involved [9]. The aneurysm is spatially arranged in a multilayered stack, as observed in Figure 3b. The innermost layer is an intraluminal thrombus, which replaces the endothelium and intima, followed by a thin media with few vascular smooth muscle cells (vSMCs) and elastin, as well as an outer layer of fibrous/inflammatory adventitia [9]. The pathogenesis of the formation of an AAA is characterized by elastin degradation, vSMC death, and increased oxidative stress [9]. Development begins with the degradation of elastin within the aortic media. Elastin, which serves to protect the blood vessels, provides the vessels with elasticity, which is a vital function to support changes in blood pressure. As the elastin is destroyed, the pressure put on the vessel by flowing blood is shifted to collagen, which is further exacerbated by the presence of hypertension [10]. Since the pressure of the circulating blood in the artery is much higher than the pressure in the adventitia, the proteases released by neutrophils move, via the gradient, through the wall and further damage the extracellular matrix, thus allowing for continued aortic dilation. The increased permeability of the wall also allows for the passage of proteolytic and oxidative species, such as plasmin and elastase, which provoke vSMC detachment and death [9]. As the matrix and smooth muscle continue to degrade, arterial wall stress increases, while arterial wall thickness decreases. The increases in pressure and stress eventually overpower the thinning arterial wall, thus leading to the rupture of the aneurysm.

### 3.2. Risk Factors and Prevalence

The prevalence of AAAs (defined as >3 cm) has been estimated at 1.4% among individuals 50–84 years of age [4]. Patients with immediate family who have been diagnosed with a AAA have a 20% chance of developing a AAA [11,12], and more than 90% of patients with a AAA have a history of smoking cigarettes [13].

While rare genetic disorders, such as Ehlers-Danlos and Marfan syndromes, may predispose patients to vascular pathology, AAAs are rarely reported in patients with Marfan syndrome [14] and were found in only 7 of 86 (8.1%) patients diagnosed with Ehlers-Danlos syndrome in a multicenter study [15]. Excluding the aforementioned genetic disorders, patient age is the single most significant predictor of AAA development [4,16]. Compared to patients under 55 years of age, individuals between 55–59 years of age have an increased risk of developing an AAA (odds ratio (OR), 2.76), steadily increasing to an odds ratio of 28.37 for individuals between 80–84 [16]. While AAAs more commonly occur in men (OR, 5.7), women carry an increased risk of rupture [17,18,19].

Smoking is the most significant environmental factor affecting both development and rupture of AAAs. Individuals with a history of smoking less than half a pack per year for at least one decade carry an increased risk (OR, 2.6), while individuals with a history of smoking more than one pack per day for over 35 years are more than 12 times more likely to develop a AAA (OR, 12.1) [4]. With each year of smoking, an individual’s risk of developing an AAA increases by 4% [20].

Additional predisposing factors include a body mass index (BMI) over 25 kg/m^2^, high cholesterol, hypertension, CAD, PAD, cerebrovascular disease, carotid disease, and family history of AAA [16]. Factors associated with decreased risk of AAA formation include smoking cessation, a diet including fruits and vegetables or nuts more than three times per week, and physical activity at least once per week [16]. A randomized controlled trial whose study population had AAAs between 2.5–5.0 cm suggested physical activity is not associated with an increased risk of rupture [21].

### 3.3. Screening

Effective screening protocols decrease mortality due to AAA rupture [4,22,23]. Physical examination has not been shown to induce the rupture of an existing AAA and is recommended for initial screening; however, sensitivity varies widely, due to multiple factors, including the patient’s body habitus and size of the aneurysm [24]. Ultrasound is the preferred method of screening, with non-radiologist-performed ultrasound having a sensitivity above 0.975 and specificity above 0.970 [25,26]. For both preoperative planning and postoperative follow-up, contrast-enhanced computed tomography (CT) is recommended in patients with an acceptable estimated glomerular filtration rate (eGFR) [4].

The SVS recommends that men or women who are 65 to 75 years of age and have a history of tobacco use receive a one-time ultrasound screening for AAAs [4]. Additionally, the SVS recommends screening of individuals 65 to 75 years of age with a family history of AAA [4]. Screening in individuals over 75, who fall into the previously mentioned categories, is recommended by the SVS only if the patient is in good health [4]. These recommendations are more extensive than those issued by the USPSTF in 2019, whose recommendations only included the routine screening of men between the age of 65 and 75 who have previously smoked, as well as the selective screening of non-smoking men within the same age range. The USPSTF does not recommend screening of women, due to insufficient evidence [6]. 

### 3.4. Surveillance

Aneurysm diameter currently serves as the accepted predictor of both future expansion, rupture, and mortality [4]. Randomized controlled trials conducted in the U.S. and Britain suggest that surveillance is sufficient, as long as the aneurysm remains below 5.5 cm [13,27,28]. The recommended frequency of surveillance is as follows: patients with a AAA measuring 3.0–3.9 cm should undergo ultrasound imaging every three years; patients with a AAA measuring 4.0–4.9 cm should undergo ultrasound imaging every 12 months; and patients with a AAA measuring 5.0–5.4 cm should undergo ultrasound imaging every six months [29]. Patients with a AAA measuring 5.5 cm or greater should be considered for treatment and referred to a vascular surgeon, as should symptomatic patients (back or abdominal pain) with an aortic diameter of 3.0 cm and above [4]. 

### 3.5. Treatment Options

Pharmacotherapy, for the sole purpose of preventing expansion or rupture of AAA, is not supported by evidence. However, pharmacotherapy to address contributory comorbidities is supported. Specifically, propranolol, angiotensin converting enzyme inhibitors, and beta blockers have not been shown to reduce aneurysm expansion in human trials, though one observational study found that ACE inhibitors may reduce mortality [30,31,32,33,34]. Statin therapy has shown mixed results [35,36,37]. Finally, beta blockers, lipid lowering agents, and ARBs have not been shown to decrease risk of rupture [4]. 

Surgical repair of AAAs falls into two categories: OSR or OAR and EVAR. While EVAR has been associated with lower perioperative mortality and postoperative complications [38], OSR remains a viable option for patients whose anatomy do not support endovascular procedures. This includes patients with vasculature that does not support either implantation of the stent or access to the implant site.

Eslami et al. (2015) proposed a risk prediction model based on factors including the type of surgery (EVAR vs. OAR), aneurysm size, age, gender, comorbidities, and creatinine [39]. This risk prediction model allows the vascular surgeon to stratify their patients into low, medium, high, and prohibitive risk groups. This model has been validated by the vascular quality initiative, a subsidiary of the SVS. Such classification can help both the surgeon and patient assess whether surgical intervention is in the patient’s best interest [39].

There is currently limited data comparing the post operative mortality rates of patients who have undergone OAR vs. EVAR. Meta-analyses conducted by Adriaensen et al. (2002) and Paravastu et al. (2014), however, did find that the short-term outcomes in patients undergoing EVAR were superior to those patients that underwent OAR [38,40]. Paravastu et al. (2014) and other investigators, however, found that individuals undergoing EVAR had higher reintervention rates and observed that the benefit of EVAR did not persist in long-term follow-up [40,41]. Perioperative outcomes, nevertheless, are improved with EVAR, when compared to OAR [42].

## 4. Conclusions

Modifiable risk factors likely contributed to the pathogenesis of the large AAA described in this case. Hypercholesterolemia led to atherosclerosis, which formed cholesterol plaques. The combination of cholesterol plaques and hypertension allowed for the destruction of the subendothelial elastin and extracellular matrix. This destruction thinned the arterial wall and allowed for aortic dilation and aneurysm formation as regular fluctuations in blood pressure pushed against the scleroitc arterial wall. When the pressure inside the vessel exceeded the strength of the now very thin wall, the aneurysm likely burst, causing a large intra-abdominal bleed, as evidenced by significant blood pooling in the left paracolic gutter that hardened during the postmortem fixation process.

The presence of an undocumented AAA of such magnitude emphasizes the importance of adhering to recommended screening guidelines, particularly in patients with multiple risk factors. While not all patients meet the candidate criteria for surgical intervention and pharmacotherapy remains unsupported, treatment modalities have improved in recent years, and patients should receive proper counseling regarding elective treatments.

## Figures and Tables

**Figure 1 diagnostics-12-02270-f001:**
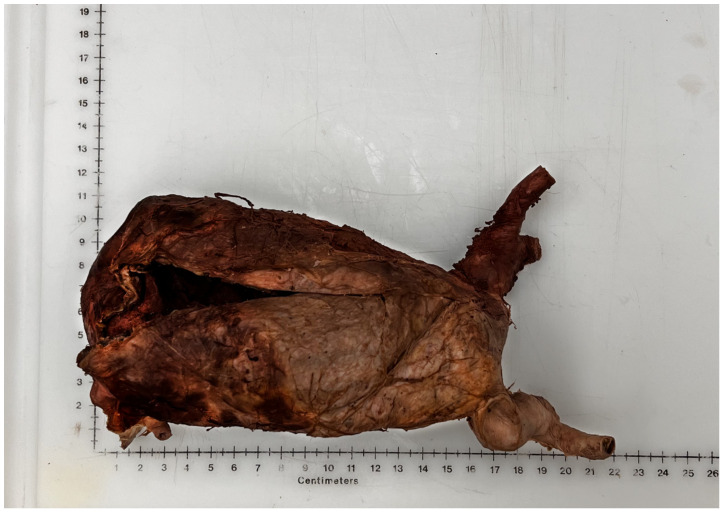
Measurement of the AAA: 8.5 cm by 17 cm.

**Figure 2 diagnostics-12-02270-f002:**
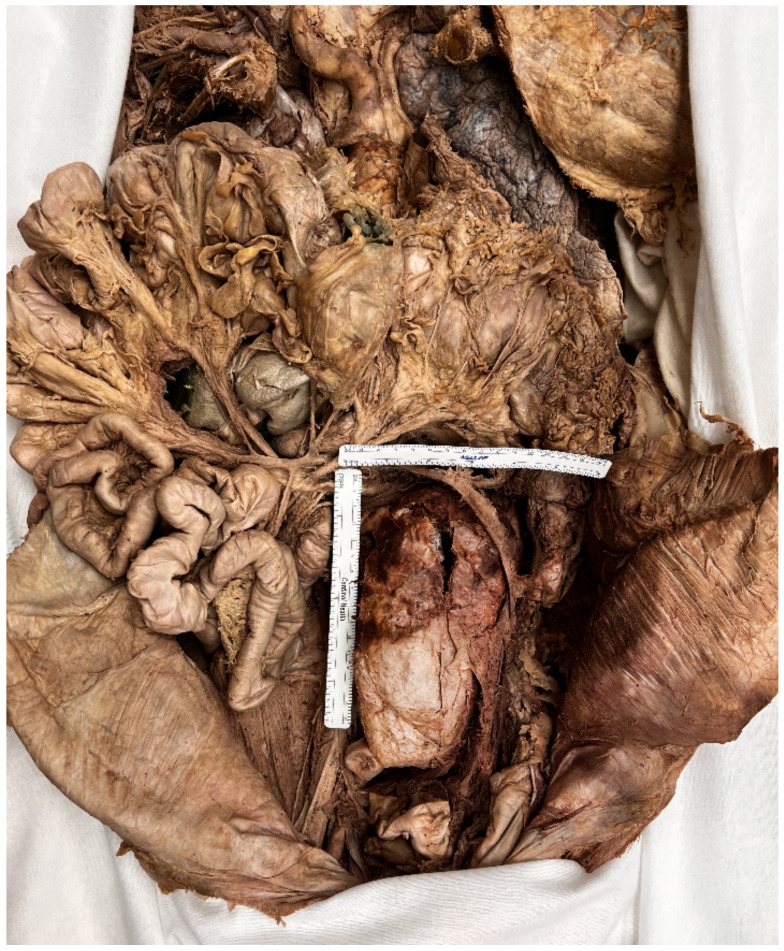
Infrarenal AAA situated above the aortic bifurcation.

**Figure 3 diagnostics-12-02270-f003:**
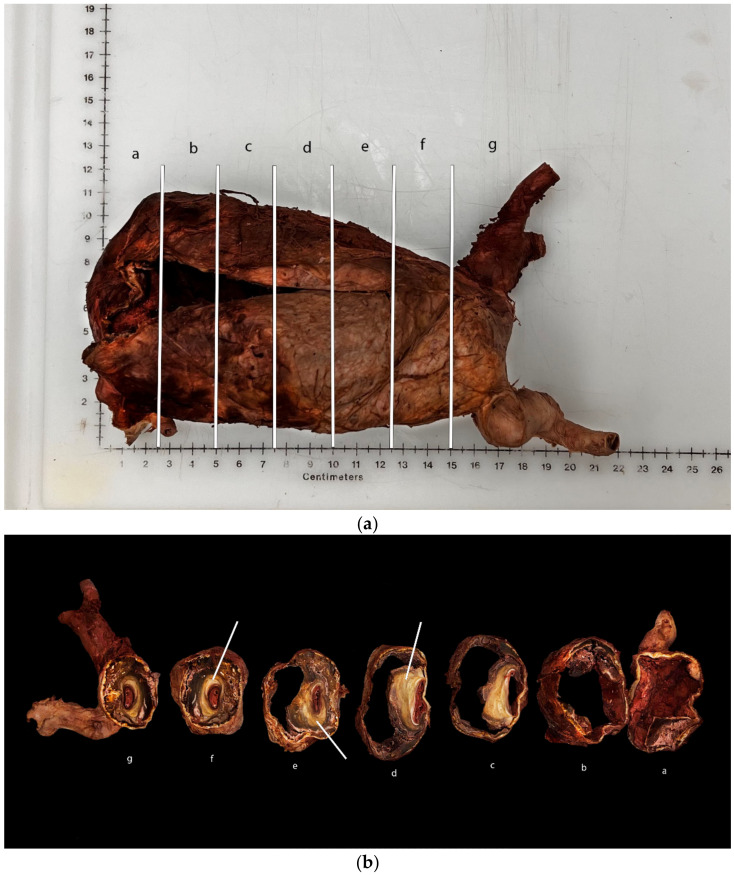
(**a**) White lines on AAA represent where the slices were made to create (**b**). Labels a–g correlate to the slices in (**b**). (**b**) Cross-sectional view of the abdominal aortic aneurysm. Layers of atherosclerosis (marked by white arrows) and calcification are observed. A small area for blood flow is marked by a red thrombus towards the center of the first four sections.

## Data Availability

No new data were created or analyzed in this study. Data sharing is not applicable to this article.

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
