# Peer review of "Chronic Atherothrombosis in a Sub-Massive Infrarenal Abdominal Aortic Aneurysm in a 91-Year-Old White Male Donor"

_diagnostics, 2022, doi:10.3390/diagnostics12102270_

Round 1

Reviewer 1 Report

Thank you for the opportunity to review your manuscript entitled "Chronic atherothrombosis in a sub-massive infrarenal abdominal aortic aneurysm in a 91-year-old white male donor".   The manuscript is well written and a stimulus for the readership.  
Red cell parameters such as RBC count and red cell distribution width (RDW)  are recognized risk factors for cardiovascular disease. A significant correlation was also demonstrated between the frailty syndrome and the RDW level (1).   Reference: 1.  Doi: 10.5603/KP.2018.0076.     Minor revisions:   1.   In the presented case, are the levels of blood morphotic parameters, such as hemoglobin level, number of erythrocytes or RDW known before death?

2. Please add the following reference:

Doi: 10.5603/KP.2018.0076.

Author Response

Sir or Ma'am,

Thank you very much for your feedback on our submission! We appreciate the time and expertise required to provide feedback on submissions of this nature and thank you for your willingness to support our research through your review.

Point by point response:

1.   In the presented case, are the levels of blood morphotic parameters, such as hemoglobin level, number of erythrocytes or RDW known before death?

1) We have very limited historical data regarding the donor. Unfortunately, the Hgb, RBC, and RDW was unknown at the time of this donor's death.

2. Please add the following reference:  

Doi: 10.5603/KP.2018.0076.

2) Thank you for recommending this article. As you stated, there is a demonstrated correlation between RBC count and mortality in patients undergoing heart valve surgery. Given the scope of the article, the authors were very careful to only include treatment data specifically regarding abdominal aortic aneurysms.

Given the unknown RBC count in this patient and the paper's specific focus on abdominal aortic aneurysms, we feel that it would be an overreach to include discussion of patient populations undergoing cardiac valve surgery.

We hope this adequately addresses your concerns with our article and once again thank you for your time and efforts in reviewing this submission.

Reviewer 2 Report

Aortic atherosclerotic lesions have been referred to in several  different ways in the medical literature. These include atheromas, protruding atheromas and atherosclerotic plaque.  This lesions can be seen in any arterial blood vessel  in the body and they are the cause of coronary  heart disease, strokes, aneurysms and of the peripheral  arterial disease.  Atherosclerosis  is a major cause of abdominal  aortic aneurysm.  Most studies  of abdominal aortic aneurysm include patients who were symptomatic and underwent  diagnostic studies. 

An abdominal aortic aneurysm is usually not a serious  health threat, but there is the risk  that a larger abdominal aortic aneurysm will rupture.  It is possible that an abdominal aortic aneurysm does not have symptoms and it tends to be diagnosed either as a result of screening or during a chech-up routine. The screening test is an ultrasound  scan which  allows the measurement of the abdominal aorta on a monitor examination.   

The presented  case highlights the particularities of abdominal aortic aneurysm in the pathogenesis of which  risk factors can intervene (smoking, age, hypercholesterol) and it has a silent evolution even at large sizes, as well as the important role that abdominal ultrasound plays in establishing the diagnosis. 

The presentation of such an abdominal  aortic aneurysm underlines the importance of screening guidelines. Not all patients meet the conditions for surgical intervention and as taking  into account the fact that pharmacotherapeutic  treatment has improved a lot in recent years, patients should  receive adequate counseling  related to the types of treatment. 

I strongly consider that the information presented is useful for medical practice. The article is well written and documented. I recommend it for publication. 

Author Response

Sir or Ma'am,

Thank you very much for reviewing our submission. We appreciate the time and effort the review process requires and thank you for taking the time to provide feedback.

Point 1:

Aortic atherosclerotic lesions have been referred to in several  different ways in the medical literature. These include atheromas, protruding atheromas and atherosclerotic plaque.  This lesions can be seen in any arterial blood vessel  in the body and they are the cause of coronary  heart disease, strokes, aneurysms and of the peripheral  arterial disease.  Atherosclerosis  is a major cause of abdominal  aortic aneurysm.  Most studies  of abdominal aortic aneurysm include patients who were symptomatic and underwent  diagnostic studies. 

An abdominal aortic aneurysm is usually not a serious  health threat, but there is the risk  that a larger abdominal aortic aneurysm will rupture.  It is possible that an abdominal aortic aneurysm does not have symptoms and it tends to be diagnosed either as a result of screening or during a chech-up routine. The screening test is an ultrasound  scan which  allows the measurement of the abdominal aorta on a monitor examination.   

The presented  case highlights the particularities of abdominal aortic aneurysm in the pathogenesis of which  risk factors can intervene (smoking, age, hypercholesterol) and it has a silent evolution even at large sizes, as well as the important role that abdominal ultrasound plays in establishing the diagnosis. 

The presentation of such an abdominal  aortic aneurysm underlines the importance of screening guidelines. Not all patients meet the conditions for surgical intervention and as taking  into account the fact that pharmacotherapeutic  treatment has improved a lot in recent years, patients should  receive adequate counseling  related to the types of treatment. 

I strongly consider that the information presented is useful for medical practice. The article is well written and documented. I recommend it for publication. 

Response 1: Highlighting the pathogenesis, screening, and treatment of this and similar abdominal aortic aneurysms was our intent with this article. We are glad that  you found these to be salient points and truly appreciate your time in reviewing our submission. Thank you once again!